# From Extremely Water-Repellent Coatings to Passive Icing Protection—Principles, Limitations and Innovative Application Aspects

**Karekin D. Esmeryan**

Acoustoelectronics Laboratory, Georgi Nadjakov Institute of Solid State Physics, Bulgarian Academy of Sciences, 72 Tzarigradsko Chaussee Blvd., 1784 Sofia, Bulgaria; karekin_esmerian@abv.bg; Tel.: +359-2-979-5811

**Abstract:** The severe environmental conditions in winter seasons and/or cold climate regions cause many inconveniences in our routine daily-life, related to blocked road infrastructure, interrupted overhead telecommunication, internet and high-voltage power lines or cancelled flights due to excessive ice and snow accumulation. With the tremendous and nature-inspired development of physical, chemical and engineering sciences in the last few decades, novel strategies for passively combating the atmospheric and condensation icing have been put forward. The primary objective of this review is to reveal comprehensively the major physical mechanisms regulating the ice accretion on solid surfaces and summarize the most important scientific breakthroughs in the field of functional icephobic coatings. Following this framework, the present article introduces the most relevant concepts used to understand the incipiency of ice nuclei at solid surfaces and the pathways of water freezing, considers the criteria that a given material has to meet in order to be labelled as icephobic and clarifies the modus operandi of superhydrophobic (extremely water-repellent) coatings for passive icing protection. Finally, the limitations of existing superhydrophobic/icephobic materials, various possibilities for their unconventional practical applicability in cryobiology and some novel hybrid anti-icing systems are discussed in detail.

**Keywords:** anti-icing; icephobicity; soot; superhydrophobicity

---

## 1. Introduction

Regardless of the dominance of digital technologies in 21st century and the increasingly computerized lifestyle of human race, sometimes reaching worrying manifestations of replacing the beauty of real-life with an imaginary virtual reality, the most meaningful, useful and efficient inventions are directly inspired by the beloved Mother Nature. Many instances of its magnificence and perfection can be given, but likely the most glaring examples of the subtle and indestructible interrelation between man and nature are the discovery of airplane, submarine, bullet train, sonar systems, cat eye-based road signs, etc. [1].

In the 30s–40s of 20th century, the nature's informative, supportive and inspirational function along with the curiosity of human mind has led to the establishment of the first theoretical principles of wetting phenomena by Cassie-Baxter and Wenzel, in context of the fast evolving textile industry [2,3]. Perhaps, at that time no one has clearly realized that the theory of wetting could be of fundamental importance for future life-changing inventions. Indeed, a few decades later, the research groups of Tomohiro Onda, Wilhelm Barthlott and Christoph Neinhuis have independently from one another marked the beginning of new era in the fabrication of super-nonwetting materials, preceded by systematic studies on the ability of sacred Lotus leaf, depicted in Figure 1, to repel rain droplets and remain perfectly dry and clean [4,5].

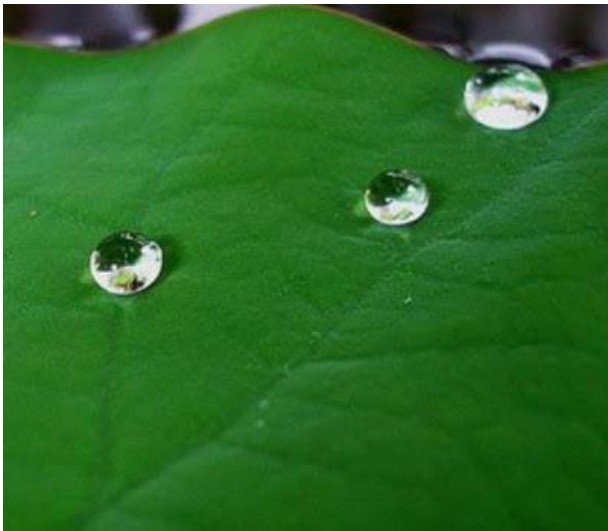

**Figure 1.** An optical image of water droplets resting on a water repellent Lotus leaf. The image is reproduced from Google under the terms of open access.

Nowadays, the contrivance of coatings (surfaces) that are capable of maintaining very weak contact with a variety of liquids such as water, oil, blood plasma, alcohol, human semen, etc. (see Figure 2), known in the scientific literature as superhydrophobic, superomniphobic or super-nonwetting depending on the liquids involved [6–10], opens a broad range of potential practical relevance. It includes, but is not limited to self-cleaning surfaces [11–13], passive icephobic [14–16] and anti-bioadhesive coatings [17–19], systems for removal of oil contamination from water basins [20–23], anti-corrosive coatings [24–26], drag-reducing surfaces [27–30], pervaporation membranes [31,32], green engineering [33] or piezoresonance chemical and biological sensors [34–38].

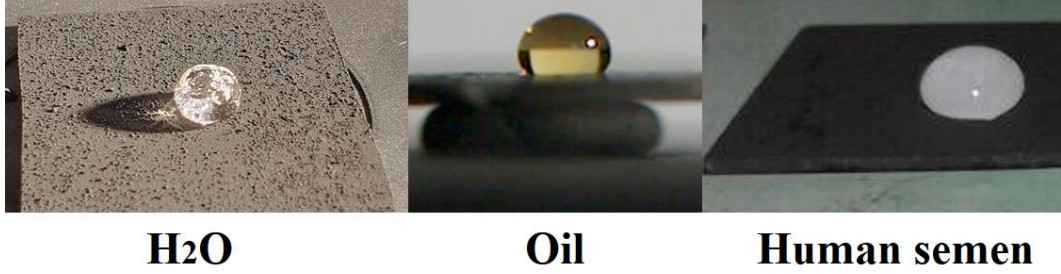

**Figure 2.** A photograph of typical super-nonwetting carbon soot coatings deposited via inexpensive and scalable flame synthesis approach [18,37,38].

Although each of the above mentioned potential real-life applications of super-nonwetting coatings has its significance and importance, the possibility for passively preventing (with no external energy applied) the ice and snow accumulation in harsh environments stands out as an approach with extremely high economic and societal impact [39–44]. The reason for such a statement is simple and associated with the detrimental effect of icing on many industrial and social sectors including aviation [42], renewable energy production [44], power and chemical plants [45,46], road infrastructure [41,43], overhead lines [47], solar panels, off-shore oil platforms [44] and many more. Virtually, the icing is undesirable literally everywhere in our daily routine, except may be in the restaurant industry, where the ice cubes are used to cool the beverages. Unavoidably, the cold weather and negative temperatures during the winter months and in some remote geographic areas lead to snow and ice accretion, and even a thin film of ice/snow is enough to alter the aerodynamic profile of wind turbines (the initially designed shape and roughness), resulting in power production losses, mechanical or electrical failure [48,49]. Furthermore, if at high altitudes an aircraft encounters cumuliform or stratiform clouds (formed upon

humid air rising through cooler surrounding air or horizontal spreading of lifted air, respectively), the supercooled water droplets occupying these clouds attach to and freeze instantly on the wings, airfoils and propellers of the vehicle [50]. In turn, excessive vibrations and increased aerodynamic drag may occur [50], regrettably in some cases leading to fatal accidents [51]. In addition, the heat exchangers of power and chemical plants located at arid regions may face freezing risks due to the excessive cooling capacity of ambient air, inflicting damage of the finned tube bundles [45]. Last but not least, the accumulation of large amounts of snow or ice on the overhead lines increases their overall mass and is very likely to destroy them, leaving hundreds of people with no telecommunications, internet or electricity [47]. In fact, such a disaster caused by freezing rain has recently been registered in some European countries, e.g., Bulgaria (cities of Belogradchik, Dobrich and Silistra in 2013 and 2018, respectively) and Russia (Moscow city, 25–26 December 2010 [52]). So obviously, finding reliable and effective solutions for mitigating or completely impeding the ice adhesion on solid surfaces is an ongoing challenge of substantial industrial and scientific value.

This review article intends to familiarize the reader with the most recent advances in the field of icephobic/anti-icing coatings. For that purpose, critical overview of the most widely spread de-icing strategies is performed, followed by concise representation of the main forms of ice and criteria for passive icephobicity. Based on this, the underlying physics of icing is considered in light of the classical nucleation theory, three-dimensional model for heterogeneous nucleation, as well as the mechanisms of vapor pressure gradient and latent heat release, closely related to the ice-bridging phenomenon and frost wave propagation. Later, the main hypotheses for the anti-icing properties of superhydrophobic/extremely water-repellent surfaces along with their major shortcomings are discussed. Finally, the launch of novel hybrid anti-icing systems and some unknown so far application aspects of icephobic materials in cryobiology are revealed.

## 2. Active and Passive Anti-icing/De-icing Methods—Critical Overview

As part of the academic community, each of us is attending international scientific conferences abroad, which usually happens by means of an airplane. If one has sit on the outer side of the airplane, next to the window, he/she might has seen occasionally that at certain atmospheric conditions passing through the clouds is accompanied by the formation of a thin shiny ice layer on the aircraft's wings (and not only). Such a layer normally disappears within a short timeframe (a few minutes) due to the activation of an integrated electro-thermal de-icing system. The electro-thermal systems are the most commonly used de-icing tools in aerospace industry and wind turbines, since they do not adversely modify the airfoil surface and do not increase the stall speed [49,53–56]. However, the optimization of these systems is still under extensive studies mainly due to the existence of several shortcomings [57]. For instance, the cyclic operation to reduce energy consumption allows accumulation of 6–7 mm thick ice that may increase the aerodynamic drag prior to removal [58]. Moreover, the electric heaters are bulky and cover only the leading edge of the surface, creating opportunities for "runback ice" accretion [58]. Furthermore, electro-thermal systems require in general materials with high thermal conductivity, which is inapplicable to the new generation erosion resistant polymers [58]. Another option for active icing protection is via mechanical vibrations excited through ultrasonic, piezoelectric or dielectric actuators [59–62]. Despite of consuming less energy, the efficiency of mechanical approach is highly restricted due to insurmountable inherent energy losses in the actuator, reducing the amplitude of mechanical oscillations applied to destroy the ice bonds.

Alternatively, the use of chemical reagents [63–65], slippery liquid-infused porous surfaces (SLIPS) [66–68] or magnetic slippery icephobic surfaces [69,70] are additional anti-icing strategies having the advantage of being passive, low weight and inexpensive techniques. Nevertheless, the chemicals utilized for road infrastructure de-icing often corrode the asphalt and generate many gaps and bumps endangering the safety of passengers [63,65], whereas the SLIPS are limited by how long the infused liquid will stay within the pores of the solid without leaking/evaporating [67]. In addition, the potential

implementation of magnetic anti-icing surfaces in practice is questionable, since their usefulness depends on the magnetic saturation of the chosen ferrofluid [69,70].

Based on the current knowledge in the field, annually, millions of euro are allocated for the funding of basic and applied research projects related to the development of sophisticated anti-icing approaches that must combine effectiveness and economical expediency (e.g., ON Wing ice detection and monitoring system, funded by EU within FP7—Transport; ref. No. 233838). Therefore, the attention of both scientists and funding agencies gradually shifts towards a relatively new concept (intensively explored for about a decade) for passive icing protection, namely using superhydrophobic coatings. The latter are prone to fair criticism, because of their fragility and lack of scalability [71], but these significant drawbacks seem to steadily being overcome and many researchers report on the fabrication of scalable, robust and wear-resistant water repellent materials [72–76].

## 3. Nomenclature of Icing and Criteria for Passive Icephobicity of Solid Materials

The nomenclature of icing is well-documented in the scientific literature [56,58,77], so here will be outlined only the most important characteristics of icing with an aim to make smooth transition into the physical mechanisms governing the ice nucleation.

Depending on the ambient temperature, pressure and formation processes, the icing can be classified as atmospheric or condensation [56,58,77]. The first type is a consequence of wet snow precipitation and freezing rain (liquid droplets with temperature below the freezing point of water i.e., supercooled droplets) or when supercooled droplets in the clouds or fog freeze immediately after impact with a solid object (e.g., an aircraft). Taking into account the density of ice and surrounding temperature, three ice regimes can be recognized: *glaze ice*—transparent, hard and the most dangerous type of ice (it completely changes the aerodynamic shape of the airfoil) obtained at temperatures up to −14 °C; *rime ice*—milky-white, feather-like ice formed at temperatures below −14 °C and following the shape of the hosting substrate; *mixed ice*—an intermediate product derived when ice particles are embedded in the glaze at temperatures around −10 ÷ −15 °C. On the other hand, the second type of icing is favorable in cold humid environments if the temperature of solid surface is below the dew point and triple point of water [78]. Such conditions promote the inception of sparse dendritic crystal structures, also known as frost [77], nucleating from a vapor phase via condensation or desublimation if both the surface and dew point temperature are below 0 °C [78,79].

Logically, if a given functional material pretends to be icephobic, it has to be capable of mitigating and/or suppressing all types of icing. From that starting point, the typical icephobic coatings must resist to the freezing of the impacting supercooled droplets (repel them or delay their freezing time) [52,58,80–82], lower the freezing temperature of water droplets at continuous solid-liquid contact i.e., sessile droplet mode (freezing temperature depression) [58,77,83] and minimizing or avoiding the frost growth [78,84–87]. However, if by some reason icing occurs, another characteristic feature indicating passive icephobicity is the reduced ice adhesion strength compared to plain materials [39,88–90].

At present, the appropriateness of superhydrophobicity for anti-icing applications is still controversial and there are extensive scientific debates on whether the extreme water repellency has something in common with the icephobicity [77,91]. This is a question of fundamental significance, which will be addressed thoroughly later in this review article.

## 4. Physics of Icing

### 4.1. Fundamental Aspects

Essentially, the freezing of water is nothing more than a first order liquid-solid phase transition triggered via nucleation and growth of the nucleus of the new phase into a three-dimensional ice crystal [92]. According to the classical nucleation theory, in order to activate the process of nucleation, a certain energy barrier needs to be overcome [92]. In the absence of solid surface (i.e., homogeneous

nucleation), the incipient nucleus is treated as a macroscopic phase of one fluid nucleating in the bulk of another phase, so the shape of nucleus is approximated to a sphere [92]. Hence, the phase transition is related to the free energy involved in the creation of a sphere of the new phase with radius $R$:

$$\Delta F = -\frac{4\pi}{3}R^3\rho_n\Delta\mu + 4\pi R^2\gamma \tag{1}$$

where $\rho_n$ is the number density of nucleating phase; $\Delta\mu$—the chemical potential of the forming phase (ice) minus the chemical potential of the nucleating phase (water); $\gamma$—the interfacial tension among two phases. The maximum free energy at the upper boundary of the thermodynamic barrier can be set as:

$$\Delta F_{HOMO} = \frac{16\pi\gamma^3}{3(\rho_n\Delta\mu)^2} \tag{2}$$

The homogeneous nucleation, described by Equations (1) and (2), occurs only for a nucleus with critical radius $R^*$ (the minimum size a group of atoms/molecules of the nucleating phase must reach prior to the initiation of phase transformation):

$$R^* = \frac{2\gamma}{\rho_n\Delta\mu} \tag{3}$$

It is pertinent to share that in most cases except in the bulk, nucleation can commence upon contact of the nucleus with the surface of a solid object, which in fact is the prevalent real-life scenario [92]. The nucleus-surface interactions are known as heterogeneous nucleation, characterized with drastically increased nucleation rate at the surface and reduced thermodynamic barrier. Here, the wettability of the surface controls the rate of phase transitions and if one analyzes the simplest case of a smooth and uniform infinite plane, the energy barrier can be expressed as follows:

$$\Delta F_{FLAT} = \Delta F_{HOMO}f(\theta) \tag{4}$$

where $\theta$ accounts for the contact angle formed among the bulk of nucleating phase (for the case of ice crystallization this is the liquid water) and the solid surface at the contact interface. Equation (4) is fundamental, as it interconnects the surface wettability with the probability for heterogeneous nucleation i.e., the likelihood for icing on solids if the considerations are restricted to water freezing. For instance, when the intermolecular attraction forces within the nucleating phase are weaker than the attraction forces of the surface itself, the value of $\theta$ is low (e.g., $\theta < 90°$) and the forming nucleus tends to spread and maximize the contact area. In contrast, if the intermolecular attraction forces within the nucleating phase are dominant, they compactly hold all its molecules and "coerce" the nucleus to "escape" from the surface, yielding high value of $\theta$ (e.g., $\theta > 90°$). Apparently, for $\theta$ approaching $0°$, virtually there is no energy barrier and the nucleation should be instant, whereas at $\theta \sim 180°$ the lack of surface contact is equalizing the barriers of heterogeneous and homogeneous nucleation and hence significantly delaying the rate of phase transformations.

Albeit very powerful, the classical nucleation theory provides solely the scaffold of the physics of nucleation and does not reflect completely the influence of surface topography and morphology on the entire process. A step forward is the development of the kinetic model for three-dimensional heterogeneous nucleation that inserts a few more parameters in Equation (4) [93]:

$$\Delta F_{HETERO} = \Delta F_{HOMO}f(m, x) \tag{5}$$

where

$$f(m, x) = \frac{1}{2} + \frac{1}{2}\left(\frac{1 - mx}{w}\right)^3 + \frac{1}{2}x^3\left[2 - 3\left(\frac{x - m}{w}\right) + \left(\frac{x - m}{w}\right)^3\right] + \frac{3}{2}mx^2\left(\frac{x - m}{w} - 1\right) \tag{6}$$

The index *m* matches θ from Equation (4), but *x* is a dimensionless parameter presenting the ratio among the radius of a solid particle around which will be created an ice embryo (i.e., the size of individual solid particles composing a given surface) and the critical nucleus radius (see Equation (3)):

$$x = \frac{R^p}{R^*} \tag{7}$$

As seen by Equation (5), along with the surface wettability, the particle size plays crucial role during the nucleation and regulates the reduction factor of thermodynamic barrier upon switching from homogeneous to heterogeneous nucleation [93].

### 4.2. Modern Implications

Since freezing on solid surfaces occurs very frequently in nature, it is of great interest to elucidate the mechanisms of ice crystallization in a variety of practical situations, including isolated droplets [52,94], groups of droplets [95] and condensation icing [96–98] with direct benefits to the future fabrication of intrinsically icephobic materials.

A recent study of Boinovich et al. shows that two typical superhydrophobic surfaces with very identical wetting properties induce notable mismatches in the freezing statistics [52]. The authors treat aluminum and steel substrates with the same hydrophobic agent (methoxy-pentadecafluorooctyl-oxy-propil-silane) and use water droplets with fixed volume, ending up with equal wetted area on both samples:

$$S_i = \pi R^2 f r \tag{8}$$

where $R$, $f$ and $r$ are the droplet radius, solid fraction in contact with the liquid and the roughness factor of the surface, respectively. Under such experimental conditions, the differences in freezing time are attributed to the availability of surface-active nanoparticles located at the droplet-air interface [52]. These nanoparticles enter the bulk of the droplet by mechanical detachment from the superhydrophobic coating and act as external centers initiating "local" heterogeneous nucleation at the liquid-air interface simultaneously with the processes at the three-phase contact line. Thus, the findings of Boinovich et al. disclose that the principles of classical nucleation theory (reducing the icing probability by simply enhancing the non-wettability of the surface) would amend in the presence of impurities [52]. If the latter accelerates the nucleation rate and the eventual freezing is unavoidable, it seems sensible to examine the complex interactions of rapidly solidifying liquid with the surface beneath it [94]. In a separate research, Graeber et al. divulge that the beginning of the freezing event at the free surface of sessile droplets (which has merit in practice, as reported in ref. [52]) leads to concentric inward motion of the freezing boundary from the air-liquid interface towards the unsolidified liquid-solid contact area [94]. As a result, the volumetric expansion of the droplet, associated with its solidification, along with the incompressibility of the droplet core and the flow constraint imposed by the outer ice shell collaboratively displace the remained liquid core towards the substrate and lift it away from the solid surface [94]. This spectacular physical mechanism, labelled as "self-dislodging", allows inherent impediment of the ice adhesion and preservation of the surface free of ice.

Nonetheless, from practical point-of-view, the water droplets always appear collectively rather than isolated and this is especially true during condensation icing, whither often the water vapor condenses into supercooled liquid droplets at multiple places across the surface [97,98]. Then, the freezing of an individual supercooled droplet may inflict further solidification of neighboring condensates via the mechanism of released latent heat, as shown in Figure 3.

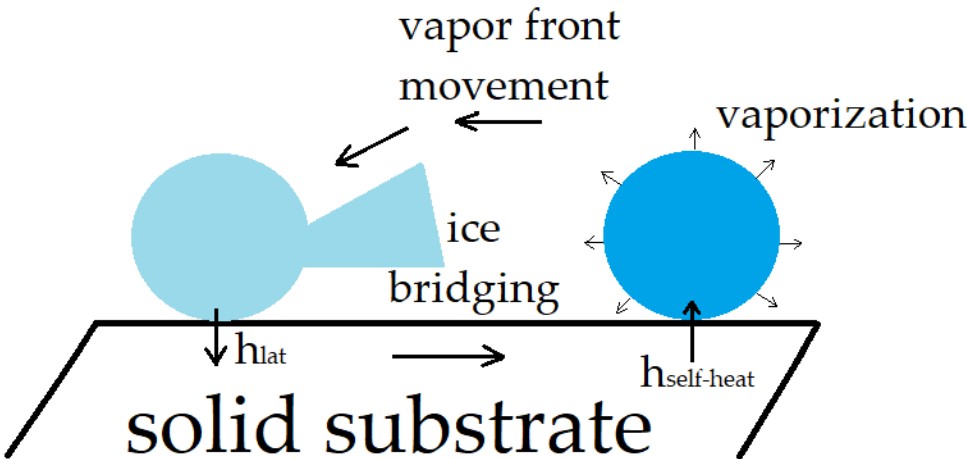

**Figure 3.** Scheme of collective droplet freezing, recreated using ref. [98].

Accordingly, the icing of a single droplet is accompanied by the generation of a latent heat flux that warms the surface and the adjacent liquid droplets begin to evaporate instantly [98]. Here, the scientific community considers two major hypotheses for the pathway of icing. First, the freely propagating vapor molecules reach the other cooled droplets and cause local supersaturation and formation of airborne ice crystals that serve as nucleation sites [95]. In turn, heterogeneous nucleation of the surrounding liquid condensates occurs via physical contact with the airborne crystals resulting in a cascade freezing [95]. Second, the water vapor attach to the frozen droplet and start the growth of ice bridges directed to the vaporizing droplet, which freezes once the bridge connects to it [98]. Interestingly, the velocity of ice bridging seems to be independent of the substrate's thermal conductivity (i.e., heat transfer rate) [98], hinting for the ascendancy of distinct freezing mechanism.

As mentioned by Chavan et al. [98], the role of latent heat is negligible and the ice bridging is primarily governed by vapor pressure gradients, whose physics is explicitly illustrated by Nath and Boreyko [97]. They explain the in-plane exchange of vapor molecules between an ice droplet and a supercooled liquid droplet with similar dimensions at constant temperature with the higher equilibrium vapor pressure of the liquefied droplet compared to the ice [97]. Thereby, the latter is regarded as a "humidity sink" harvesting vapor from the neighboring droplets and creating ice bridges that are transferred to the next row of cooled condensates in the form of a chain reaction [97].

Inspite of a couple of uncertainties in the presumed phenomena responsible for the freezing of water droplets (e.g., cascade freezing documented entirely in low-pressure environments [95]; irrelevance of the latent heat to the droplet evaporation established predominantly by computer simulations [98]), all of the proposed freezing routes are fundamental and sufficiently reliable for description of the interfacial processes throughout icing.

## 5. Modus Operandi of Superhydrophobic Coatings for Passive Icing Protection

One of the first attempts for integrating the hydrophobic effect in ice releasing coatings are documented in the early 00s of the 21st century as spontaneous desire to interrelate the collected at that time scientific knowledge for the fundamentals of non-wettable plants with the chance to avert the adherence of ice on solid surfaces [99–101]. Considering concisely the wetting theory of Cassie-Baxter, described comprehensively elsewhere [102,103], the physical micro- nano-roughening of a given hydrophobic surface yields a solid-liquid interface at which the water droplet no longer retains complete contact with the solid at all interfacial points. In this extreme case, the inability of the liquid to penetrate the spacing among surface features leads to an apparent contact angle determined with the following equation:

$$\cos\theta_{CB} = f_1\cos\theta_\gamma - f_2 \tag{9}$$

where $\cos \theta_\gamma$ is the intrinsic contact angle for a smooth hydrophobic solid, while $f_1$ and $f_2$ reveal the total area of the solid-liquid and liquid-vapor interfaces, respectively. Since $f_1$ depends on the surface topography, the mathematical equation fully addressing the non-wetting regime is:

$$\cos \theta_{CB} = r_1 f_s \cos \theta_\gamma - (1 - f_s) \tag{10}$$

where $r_1$ is the roughness factor and $1 - f_s$ is the area bridged across the surface protrusions. As a result of the smaller contact area, the solid-liquid attraction is weakened, the effect of hydrophobicity is amplified and the attached water droplet easily rolls along the solid surface.

So, the basic idea for the icephobicity of superhydrophobic coatings, provoked by their "slippery" Cassie-Baxter state, relies on the nearly perfect transfusion (with negligible losses) of the kinetic energy of impinging supercooled water droplets into surface energy and vice versa [104]. Such an effect emanates from the virtually frictionless nature of all superhydrophobic materials due to the presence of interfacial slip [16,102]. The latter prevents the substantial kinetic energy dissipation that is normally triggered by the surface friction and liquid's viscosity, and creates repulsive forces within the droplet. Hereby, in freezing rain conditions, the impacting water droplets bounce-off the surface immediately, which decreases the overall solid-liquid contact time down to a few milliseconds, lessens the exchanged thermal energy below the critical threshold needed to trigger ice nucleation and the water is extricated prior to freezing [105]. Other benefit of superhydrophobicity is related to the minimized solid-liquid interfacial area retarding the heat transfer rate and reducing the icing probability, since both depend on the surface area and wetting state [80,106–109]. An attractive analogy of the underlying physics of this effect, intended for non-specialists, is a fleshpot with soup mounted on a hot plate, assuming that the fleshpot's and hot plate's diameters match and the heating temperature and soup's volume are constant. The time required to warm up the food will depend on the amount of heat transferred by the hot plate per unit time and unit surface area, divided by the existing thermal difference (e.g., a colder fleshpot will warm longer than a warmer one—at fixed heating temperature). Now, if one modifies the hot plate in a way ensuring highly rough surface profile, the absolute number of solid-solid (plate-pot) contact points will decrease at the expense of newly appeared empty spaces i.e., air gaps. In such a complex system, the heat will reach the pot by two routes—through the solid surface area in intimate contact with the pot and via the air gaps. Since the heat propagates much slower through the air, compared to conductive solids, the overall heating time will increase proportionally. While this is a simple analogy, the conception of freezing time delay due to limited solid-liquid interfacial area is valuable when designing passive icephobic materials. Importantly, even if icing event takes place, the retained air inflicts stress concentrations at the contact interface and impairs the ice bonds [58,84], resulting in up to 80%–90% lower energy necessary to maintain the protected surface free of ice [110,111]. Furthermore, the non-wetting Cassie-Baxter regime intrinsic for any superhydrophobic coating may support the freezing of droplets in a spherical shape (the so-called "ball-up" freezing), exemplified in Figure 4, and their facile dynamic detachment with low amount of applied thermal energy [105,112–114].

Currently, the control and manipulation of ice nucleation and accretion by means of hydrophobic surface modifications is a subject of extremely active ongoing research [115–133]. Some of the most significant scientific achievements up to now indicate encouraging opportunities to decelerate the freezing process with more than 20 min [122,132] and impressively even for hours [52]. Furthermore, a few cutting-edge approaches for the fabrication of icephobic coatings clearly prove that the freezing of sessile or impacting water droplets could be shifted below −15, −20 [116,119,130] and even −35 °C [105]. Moreover, a very recent and highly innovative study suggests a passive technique for ice wicking [115] and suppression of condensation icing on overhead cables [126] via tuning the surface wettability/orientation and adding an array of hygroscopic micro-groove rings that foster overlapping dry zones, free of dew and frost for indefinite time.

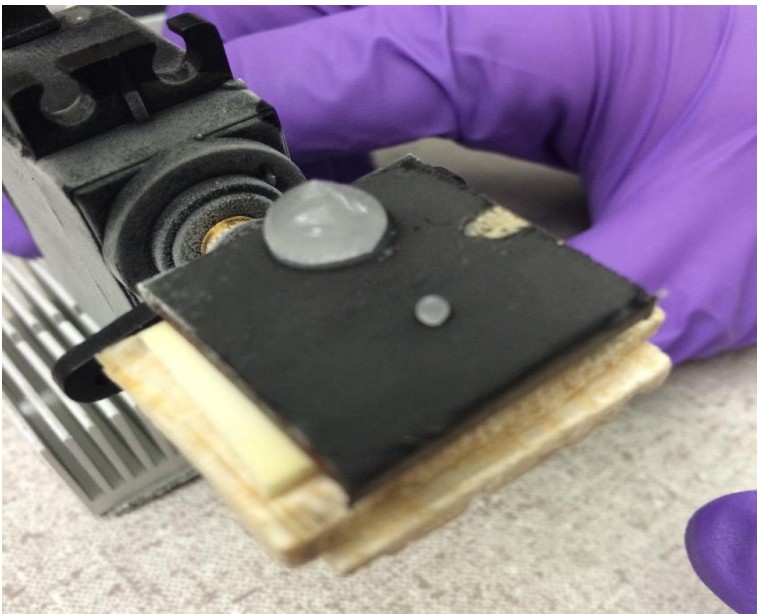

**Figure 4.** A photograph of "ball-up" freezing on an icephobic carbon soot coated aluminum substrate with surface temperature of ~−10 °C, exposed to room temperature conditions (*T* ~22 °C; RH—50%).

Unfortunately, the research in the last decade still fail in answering unambiguously one fundamental question, namely are the superhydrophobic materials indeed icephobic and if not why. Most of the eminent scientists share similar persuasion that superhydrophobicity and icephobicity are terms with different meaning [58,77,91,134–136], although the newest scientific evidences support the assumption that direct correlation between water and ice repellency is visible [131]. Four major and solid arguments defend the theses for diverse physical basis of superhydrophobicity and icephobicity in terms of performance and properties. First, not every superhydrophobic surface (coating) may be classified as ice repellent, because the critical size of the solid particles that compose any surface is in different scales for superhydrophobicity and icephobicity, and as shown elsewhere, above a certain particle diameter the ice accumulation is favorable [137]. Second, the forces needful to detach a liquid water droplet are associated with the contact angle hysteresis (low hysteresis accounts for "slippery" Cassie-Baxter wetting state [102]), whilst a piece of ice would be pull away under specific configuration of interfacial cracks [135]. Third, the roughness factor strongly affects the icephobicity of materials and the surfaces with nanometer-scale roughness manifest lower ice adhesion compared to the highly rough counterparts (e.g., micron-scale roughness) [91,138]. Forth, the anti-icing efficiency of superhydrophobic coatings is limited at multiple icing/de-icing cycles due to mechanical damage of surface features, as well as in cold humid atmosphere due to eventual frost formation, loss of extreme non-wettability and increased ice adhesion strength [136]. Luckily, science circumvented the last obstacle and now functional anti-frosting/icephobic surfaces with satisfactory mechanical robustness are readily available [78,84–87,105,126,139]. Preserving the surface free of frost in humid environments could be accomplished via precise adjustment of the spatial distribution of the adjacent nucleation (hydrophilic) sites and hence, retardation of the ice bridging and interdroplet frost wave velocity [78,85,87,140], summarized illustratively in Figure 5.

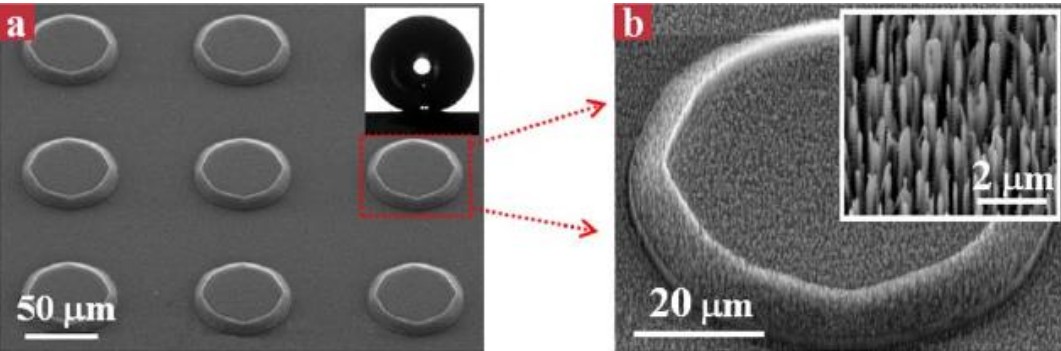

**Figure 5.** Condensation frosting on hierarchical nanograssed superhydrophobic surfaces. (**a**) Top view scanning electron micrograph and (**b**) close-up image of a single nanograssed micro-truncated cone. The image is reproduced from ref. [96] under CC BY 3.0 license for unrestricted use, redistribution and modification of published research. In principle, the frost spreading is successfully accelerated or decelerated by inserting or removing the active sites (in Figure 5 approximated with each nanograssed cone) [87] and deliberately triggering an early freezing event to minimize the size of incipient nuclei [78] i.e., via appropriate laboratory micropatterning.

Additional drawback for undisputedly correlating superhydrophobicity with icephobicity is the elusive influence of the non-wettable film's thickness and air cushion (i.e., plastron) convection on its anti-icing performance [141,142]. In particular, increasing the thickness of the protective coating is believed to enhance the thermal resistance and reduce the heat transfer to a given sessile droplet, monotonically elongating the freezing time [141]. Such an observation is indicative for the paramount role of film thickness, since it also determines the level of mechanical stresses experienced by the blades of wind turbines, for example, and the fatigue life of the material [143]. Curiously, the thermodynamic free energy barrier and icing probability of water droplets depend in addition on the dynamic motion of air cushion, formed by the air pockets confined within the surface protrusions, i.e., the exchange of thermal energy between warmer air beneath the droplet and colder surrounding air during the cooling process [142]. Therefore, the optimal design of passive icephobic surfaces must comprise suitable balance between the coating's thickness and the effects of air convection, which at the moment are largely ignored [141,142].

The personal view of the author of this review, without pretending to be correct or exhaustive, is that superhydrophobicity and icephobicity are terms with equal physicochemical origin, since water and ice are two distinct physical states of one chemical substance. A fair parallel would be the scientific nomenclature of wetting, according to which any surface with static water contact angle (SCA) above 150° and contact angle hysteresis below 5° possesses superhydrophobic properties with extreme droplet mobility [102]. In terms of the wetted area, however, the solid-liquid contact interface of a surface with SCA = 170° is smaller compared to a counterpart with SCA = 150°. Therefore, at equal other conditions, the associated heat transfer rate proportionally decreases when switching from the pattern with SCA = 150° to the one with SCA = 170°, although both samples are labelled as superhydrophobic. In other words, one can try artificially to separate these surfaces to "more super-nonwettable" or "less super-nonwettable", ignoring the nomenclature of wetting, where such segregation is not very common. Saying that, the superhydrophobicity and icephobicity might be undisputedly related, because both superhydrophobic surfaces with particle size within and beyond the range of icephobicity provide apparently better anti-icing performance than the hydrophilic doublets, especially at outdoor tests [137], as shown also in Figure 6.

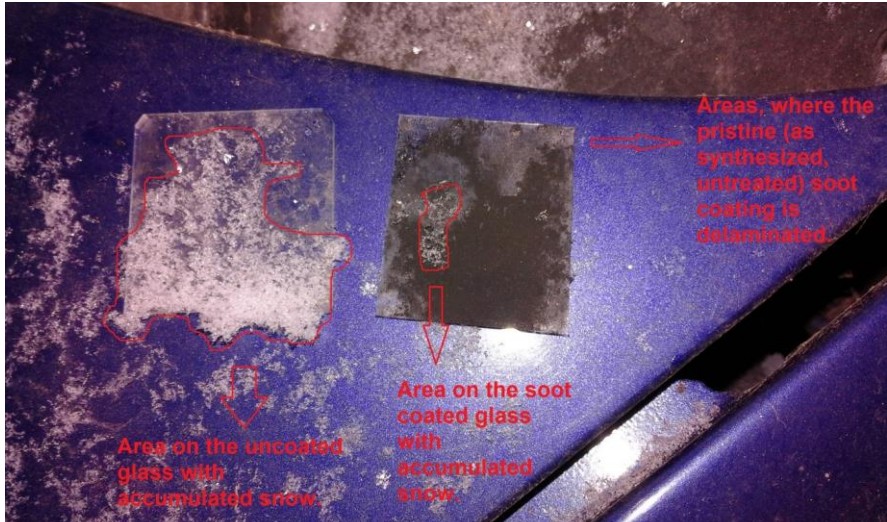

**Figure 6.** Snow and ice accumulation on an uncoated (left side) and soot coated (right side) glass slide upon prolonged 7 days exposure to real-life winter conditions ($T \sim -2 \div -5\ °C$; RH—60%–80%). The experiments are performed at Scientific Complex 2 of Bulgarian Academy of Sciences, Sofia, Bulgaria in January 2018. As seen, about 75% of the surface area of the uncoated glass is covered with snow/ice, whereas not more than 5% is the region with attached snow/ice on the soot coating.

## 6. Novel Hybrid Anti-Icing Systems

An objective and bias-free analysis of the state-of-the-art in anti-icing materials employing the virtue of water repulsion, proffers that functional material alone may not be able to ensure the desired efficacy for aircraft or wind turbine de-icing operations [144,145]. Instead, combining an icephobic coating with electric heating emerges as a convenient option for icing mitigation in atmospheric conditions [144–150]. The focus in these hybrid systems is on the reduced power consumption up to 50%–90% both in glaze and rime ice regimes [145,148]. This is achievable by chemically tailoring the surface under test in a way ensuring weak water-solid interactions, while in the meantime the heating element is covering only the leading edge of the surface. In turn, the ice accretion in the hydrophobic surface area is low, due to the fast runback speed of the impacting water droplets, whilst thermal energy is exploited predominantly for eliminating the attached ice at the leading edge [145]. Even though the hybrid strategy is promising for real-life performance, for the case of electromechanical de-icing is of utmost importance to choose a coating that would not adversely affect the stress generation, as the latter defines the coupling between the source and the ice-substrate interface [144].

## 7. Fundamentally New Potential Applications of the Passive Icephobic Coatings in Cryobiology

Even though the studies in passive icephobic coatings are mainly referred to economic, energy and safety implications [109], the freezing phenomena are likewise vital in cryobiology and medicine [151]. The conservation of living matter (both human and animal) at sub-zero temperatures is a rapidly developing scientific branch with benefits to the reproductive and regenerative medicine for supporting the retention of male/female fertility prior to gonadotoxic radiotherapy and chemotherapy or storing sperm cells throughout the mandatory screening procedures for sexually transmitted diseases prior to the artificial insemination [152]. Knowing the suppressed biochemical interactions in the living matter at cryogenic temperatures, the creation of a protective environment for freezing of cells and tissues is feasible by means of gradual (slow) or instant freezing (vitrification) [153]. A serious shortcoming of cryoconservation technology is the deteriorated vitality and structural integrity of the frozen cells/tissues caused by osmotic shocks, exposure to intracellular solutes with high concentrations or toxicity of the chemicals (cryoprotectants) utilized to decrease the freezing point of liquid suspensions and minimize the intracellular ice nucleation events that rupture the cell membrane [154–156].

Obviously, finding reliable measures for solving the above-designated problems would be highly beneficial from scientific and industrial perspective, and here is the fundamental intersection among condensed matter physics and biology. Perhaps the readers of this article are familiar with one famous American romantic comedy entitled "When Harry met Sally", where the main characters meet just before sharing a cross-country drive and then within twelve years have several accidental encounters ending up with marriage. Analogous story, although not love related, is the basis for the execution of the first of its kind experiments revealing cryoconservation of human semen by means of super-nonwetting icephobic carbon soot [157]. In March 2015, the author of this review meet Mr. Todor Chaushev, a microbiologist interested in the invention of new technologies for diagnosis and selection of human gametes and embryos. About two years later, in January 2017, both individuals discuss the anti-icing performance of the soot, in particular the ability of this material to minimize the heat transfer at the solid-liquid interface [105]. The fruitful conversation instigate one extraordinary and somehow "crazy" idea for the integration of icephobicity in cryobiology. An extensive literature survey in the databases of all reputable publishers and publishing companies (e.g., Elsevier, Clarivate Analytics, Nature, Science, American Chemical Society, Royal Society of Chemistry, Springer, American Institute of Physics, Institute of Physics, MDPI, etc.) surprisingly shows complete lack of experimental evidences for the role of heat transfer phenomena, regulated by the surface wettability, in cryoconservation of living organisms. Just one editorial report and an online communication very scarcely mention that "freezing and manipulating cells encapsulating nanoliter droplets on super-hydrophobic nano-rough surfaces is one of the potential ways of manipulating droplets, as minimal volume technologies offer potential solutions to the current challenges of cryobiology" [158,159]. Quite excited, both scientists decide to perform first laboratory trials via a simple homemade setup, illustrated in Figure 7 (further details can be found in ref. [157]).

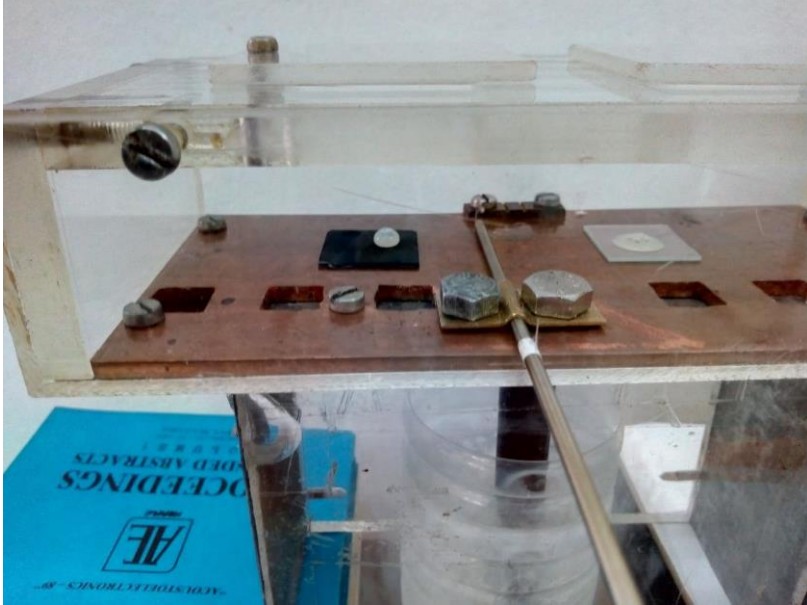

**Figure 7.** Cryoconservation of a seminal fluid through icephobic carbon soot coatings.

The freezing and thawing of sperm cells of three patients with distinct seminal characteristics (i.e., normozoospermia, oligozoospermia and asthenozoospermia) is much more successful on the icephobic soot compared to an uncoated glass substrate and about 80% of initial sperm motility is recovered [157]. Such an outcome is very unexpected and emotive, because a recent study reports not more than 70% recovered motility achieved through larger quantity of cryoprotectants in the seminal suspension [160]. Of course, notwithstanding of the positive feedback from the scientific community ("highly innovative and unconventional approach"; "an eye-opening short communication", etc.),

many important questions yet remain unanswered, one of them reminding that the proposed technique seems to contradict the basic principles of cryobiology.

Two main scientific concepts, adapted from the excellent works of Boinovich et al. [52] and Graeber et al. [94], might justify the observable improvement of cryoconservation procedure. Namely, the soot retards the heat transfer rate at the solid-liquid interface [52], ensuring slow water influx within the cells upon thawing, whilst the movement of the freezing front from the three phase contact line towards the liquid's bulk [94] inflicts outward osmosis and uniform dehydration. Definitely, the described mechanisms are prone to critical disputes due to the lack of irrefutable evidences [157], however, the exploration of something newfangled is often accompanied by a degree of uncertainty and contradiction of well-established principles in a given field. Thus, one may accept the preliminary findings as the ground floor of a big and complex building under construction.

## 8. Conclusions

Hitherto, the launch of passive anti-icing/de-icing coatings in industrial setting is yet (regrettably) at a bottleneck stage principally due to operability and scalability difficulties, as well as failure in fully understanding the contribution of surface parameters on the ice and snow accretion under natural freezing conditions. Adjusting the solid surface characteristics to gain optimal icephobic performance is obligatory, but demands future profound analysis of the impact of protective coatings' thickness and air cushion convection on the icing probability and freezing delay in order to establish universal "recipe" in designing anti-icing materials. In addition, special attention must be devoted to the mechanical durability of the as-prepared icephobic surfaces, since there is plenty of research space in improving their wear resistance in real-life environments (out-of-lab). Last but not least, the debate issues related to the link among superhydrophobicity and icephobicity need to be addressed in the foreseeable future. At the same time, the applicability of modern icephobic coatings could be expanded into one very exciting and novel direction, in particular, for conservation of living matter at cryogenic temperatures. The probable societal benefits of such a practical transformation are obvious and associated with reduced number of painful biopsies and repetitive surgeries or increased success rate of the in-vitro fertilization. It is the author's belief that "science without borders", the motto of the renowned philosopher and philanthropist Alexander von Humboldt, and "sky is the limit" are aphorisms with strong message to the humanity, indicating that "impossible" is an imaginary term, but the future life-changing inventions depend on our willingness to think analytically, being appreciative and live in resonance with Mother Nature.

**Funding:** This review article precedes future systematic experimental studies on the impact of physicochemical characteristics of super-nonwetting carbon soot coatings on their icephobic properties, funded by the Bulgarian National Science Fund under framework programme 2020–2022 (grant No. КП-06-Н37/7/06.12.2019).

**Conflicts of Interest:** The author declares no conflicts of interest.

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
