# Peer review of "From Extremely Water-Repellent Coatings to Passive Icing Protection—Principles, Limitations and Innovative Application Aspects"

_coatings, doi:10.3390/coatings10010066_

Round 1
Reviewer 1 Report
The author proposed a review of interesting field. However, the manuscript appears more like a report or an opinion than a useful review for the scientific community. The manuscript have to be reviewed deeply, and the recent achievement in the field should be reported and discussed, as a review should be. I can not recommend this review for publication.
The title is misleading and had little connection with the reported studies.
For example, the author reported the effect of cold weather in cities and real life, which is not convenient for a scientific review. This part should be removed and replaced by a serious discussion dealing with coatings.
The author proposed an interesting part of the physics of the icing phenomenon, but the physics of superhydrophobic coatings are missing: Cassie-Baxter’theory for example.
Furthermore, the authors should report the chemistry, the structure and the physics behind the superhydrophobic surfaces.
Author Response
Referee 1:
1) The author proposed a review of interesting field. However, the manuscript appears more like a report or an opinion than a useful review for the scientific community. The manuscript have to be reviewed deeply, and the recent achievement in the field should be reported and discussed, as a review should be. I can not recommend this review for publication.
Response:
Dr. Esmeryan thanks the reviewer for his/her scientific critique. In his/her prefatory words, the reviewer questions the structure of the article and claims it “appears more like a report or an opinion than a useful review for the scientific community”. Well, to address this particular comment, Dr. Esmeryan likes to directly quote the definition for a review article and then comment it in detail (https://en.wikipedia.org/wiki/Review_article):
“A review article is an article that summarizes the current state of understanding on a topic.[1] A review article surveys and summarizes previously published studies, rather than reporting new facts or analysis. Review articles are sometimes also called survey articles or, in news publishing, overview articles. Academic publications that specialize in review articles are known as review journals.
Review articles teach about:
the main people working in a field recent major advances and discoveries significant gaps in the research current debates ideas of where research might go next”Based on the above definition, the submitted manuscript fully covers all mandatory elements of one review article. First, it definitely teaches about the main people working in a given field – the article contains 159 literature sources, where the names of top experts in the field of wetting phenomena and anti-icing surfaces/coatings are clearly visible. Examples include, but are not limited to: the research groups of Prof. David Quere, Prof. Glen McHale, Prof. Sergei Kulinich, Prof. Masoud Farzaneh, Prof. Joanna Aizenberg, Prof. Michael Nosonovsky, Prof. Ludmila Boinovich, Prof. Dimos Poulikakos, Prof. Ivan Parkin, Prof. Alidad Amirfazli, Prof. Marco Marengo, Prof. Elmar Bonaccurso, Prof. H. Yildirim Elbir, Prof. Gareth McKinley, Prof. Richard Sear, Dr. Carlo Antonini, Dr. Jonathan Boreyko, Dr. Nenad Miljkovic and many, many more (the list is too long to be embedded here).
Second – the submitted review article definitely teaches about recent major advances and discoveries, and this is undisputedly evident by reading sections 1, 4.2, 5 and 6.
Third – the significant gaps in the research are clearly presented by mentioning the operability and scalability drawbacks that must be overcome in order to incorporate the passive icephobic coatings in industrial setting (please see sections 2 and 8)
Fourth – it is more than obvious than the submitted manuscript covers the current debate issues in the field of passive anti-icing materials (please see sections 3 and 5). This is very kindly acknowledged by reviewers 2 and 3 who convincingly recommend acceptance of the article.
Fifth – Dr. Esmeryan thinks it would not be necessary to comment whether the manuscript provides new ideas of where the research might go next. Simply, anyone who is still doubting could read sections 7 and 8, and he/she will be assured.
According to the official nomenclature for a review article and the above analysis, the submitted manuscript obviously meets all scientific criteria. Therefore, this particular argument of reviewer 1 could be considered as irrelevant. Moreover, counting again on the nomenclature for review articles, “An expert's opinion is valuable, but an expert's assessment of the literature can be more valuable. Readers benefit from the expert's explanation and assessment of the validity and applicability of individual studies”. In turn, the fact that Dr. Esmeryan shares his point-of-view for some physical mechanisms and provides attractive explanations of some physical phenomena could be acknowledged and welcomed rather than criticized.
2) The title is misleading and had little connection with the reported studies.
Response:
Well…. this is a solid argument and Dr. Esmeryan thanks the reviewer for this particular comment. Based on the feedback received so far, the author decided to change the title of the review article. The new title is: “From Extremely Water Repellent Coatings to Passive Icing Protection – Principles, Limitations and Innovative Application Aspects”. Dr. Esmeryan believes that this new title entirely addresses the manuscript’s content and is much more appropriate than the first one (although the first one is quite attractive and would catch the attention of the broad readership of Coatings).
Change in Manuscript:
The title of the review article has been changed.
3) For example, the author reported the effect of cold weather in cities and real life, which is not convenient for a scientific review. This part should be removed and replaced by a serious discussion dealing with coatings.
Response:
One very useful phrase from the English language can be used here, namely “Lets agree to disagree…”. What the reviewer is stating here is ok only if the main idea and focus of the review is on the structure and properties of the coatings. Fortunately, this is not the case and the aim of Dr. Esmeryan is to familiarize the readers with the problems associated with atmospheric and condensation icing, the major physical mechanisms regulating the ice incipiency on solid surfaces and summarize the breakthroughs in the field (the results themselves rather than commenting individual coatings). Furthermore, the author of the review does not understand how one can justify the importance of a particular research field if he/she is unable to clearly define the main existing shortcomings. Leaving hundreds of households with no electricity, telecommunications and internet due to freezing rain events is definitely a huge problem and there is no argument on the importance of mentioning natural disasters. Btw, just a few weeks ago (December 7th 2019), there was a massive car accident (20 cars or something like that) in Sofia Bulgaria again due to freezing rain. Moreover, the idea for mentioning “cities” is inspired from the famous and very interesting article of Boinovich et.al. (please see reference 52, page 1659 – just before the start of the last paragraph), which has received about 122 citations for the period of 5 years.
Change in Manuscript:
There is no change in the review article associated with the above comment. The justification of such a decision is provided above.
4) The author proposed an interesting part of the physics of the icing phenomenon, but the physics of superhydrophobic coatings are missing: Cassie-Baxter’theory for example.
Response:
Dr. Esmeryan is very happy that at least this part of the review article seems meaningful for the reviewer. The only reason the Cassie-Baxter’s theory of wetting has not been included in the first draft of the article is because this theory is explained very comprehensively elsewhere (please see reference 102, cited more than 400 times so far). However, after carefully analyzing the reviewer’s comments, the author decided to include a short paragraph in the text, where the theory of wetting is correlated with the icephobic performance of superhydrophobic coatings (surfaces). The addition of this paragraph enhances the quality of the review, so Dr. Esmeryan is thankful for this suggestion.
Change in Manuscript:
A new paragraph linking the theory of wetting with the anti-icing properties of superhydrophobic coatings (surfaces) is now available in Page 7, Lines 276-292 in the revised manuscript.
5) Furthermore, the authors should report the chemistry, the structure and the physics behind the superhydrophobic surfaces.
Response:
Not really – very detailed and useful information about the chemistry, structure and physics of superhydrophobic surfaces is readily available in many review articles. Examples – Shirtcliffe et.al. An introduction to superhydrophobicity, Adv. Colloid Interface Sci. (2010); Sojoudi et.al. Durable and scalable icephobic surfaces: Similarities and distinctions from superhydrophobic surfaces, Soft Matter (2016); T. Xiang et.al. Superhydrophobic civil engineering materials: A review from recent developments, Coatings (2019); Y. Lin et.al. Recent progress in preparation and anti-icing applications of superhydrophobic coatings, Coatings (2018); N. Namdari et.al., Advanced functional surfaces through controlled damage and instabilities, Mater. Hor. (2019); Gh. B. Darband et.al. Science and engineering of superhydrophobic surfaces: Review of corrosion resistance, chemical and mechanical stability, Arab. J. Chem. (2018); J. T. Simpson et.al., Superhydrophobic materials and coatings: A review, Rep. Prog. Phys. (2015).
Dr. Esmeryan shows in this revision cover letter 7 examples of review articles, some of them cited in the manuscript, but he is very convinced that could find at least another 10. So, the main question here is – how would the scientific community treat an article that repeats the structure and content of other articles? As a reviewer himself (invited reviewer for 20 high-profile, high-impact journals with more than 20 review invitations per year. https://publons.com/researcher/1425007/karekin-esmeryan/peer-review/), the author of the present manuscript would say – certainly negatively. Repeating other researchers’ work (structure of the articles) without adding any novelty and originality is more than clear evidence for the absolute lack of creativity. Therefore, Dr. Esmeryan wants to avoid such a bad impression and his review article has a few original aspects that would meet the publication standards of the high-quality scientific journals. In particular, there is no published review article on superhydrophobic or anti-icing materials that reveals a completely new and exciting direction of their practical application, namely in cryobiology. In addition, the description of the debate issues related to the link among superhydrophobicity and icephobicity is expanded to the case of protective coating’s thickness and air cushion (plastron) convection – two features scarcely considered in the literature up to now (please see references 140, 141). Furthermore, Dr. Esmeryan is not aware of the existence of many review articles summarizing the spectacular physical pathways of water freezing, previously established by the research groups of Prof. Poulikakos and Dr. Miljkovic – this is another original aspect of the present review article. And finally, comments regarding the ice wicking and frost-free maintenance of overhead cables, reported by the group of Dr. Boreyko, are not available in other review articles.
Even though reviewer 1 might feel irritated and recommend rejection again, Dr. Esmeryan would not change the structure of his article, since he strongly believes in its originality (so as the other two reviewers). The author of the review is convinced that writing articles in a slightly unconventional way is very beneficial and creative, and would inspire the readers to rethink the world we are living in. As scientists, we might be able to achieve huge progress in any of the fields of science, but this must not be at the expense of lost moral, honesty and integrity.
Change in Manuscript:
There is no change in the review article associated with the above comment. The justification of such a decision is provided above.
6) “Extensive editing of English language and style required”
Response:
Here, Dr. Esmeryan is highly surprised from the reviewer’s assessment. In fact, for almost 10 years in science, this is the first time when a reviewer is claiming that the English language and scientific style of the author are poor… There must be some mistake, because the other two reviewers score the style as very good (4 or 5 stars of 5 possible). Moreover, if the language and style are indeed poor and do not meet the publication standards, then, on what basis Dr. Esmeryan publishes as a first author in reputable scientific journals (e.g. Applied Surface Science, Colloids&Surfaces A, Materials&Design, MDPI Sensors, Journal of Physics D Applied Physics, etc.) and serves as an invited reviewer for 20 well-recognized scientific journals? With all his respect, but Dr. Esmeryan strongly recommends the reviewer to extract all passages from the manuscript with poor English grammar and prove his/her opinion with undisputed linguistic evidences. Moreover, the author of the review thinks that if one has critical comments regarding the style and English language, he/she must make sure that his/her writing is close to perfect. Currently, the reviewer’s report contains several errors that are readily recognizable for anyone who speaks fluent English. Nevertheless, Dr. Esmeryan has paid full attention to the English grammar and in some places throughout the text it is corrected.
Change in Manuscript:
Full attention to the English grammar has been paid and where necessary, corrected.

Reviewer 2 Report
In this review manuscript, the author aimed to reveal the relationship between icephobic and superhydrophobic coatings. Firstly, active and passive anti-icing methods were summarized and their physical mechanisms regulating the ice accretion on solid surfaces were proposed. Then the most important scientific breakthroughs in the field of icephobic coatings were reviewed. The main hypotheses for the anti-icing properties of superhydrophobic surfaces along with their major shortcomings were discussed. Finally, the launch of novel hybrid anti-icing systems and some unknown so far application aspects of icephobic materials in cryobiology were revealed. This review manuscript was well written and the author cited important and comprehensive literatures to support his opinions. I recommend it to be accepted after minor revision. Following are the comments:
The title of the article may be inappropriate, for bio-inspired science or technology were not mentioned much in the manuscript. Why did author use “soot” as key word? Looking through the manuscript, several samples were given using carbon soot, however, the author didn’t explain the functionality of soot. In page 8, line 303, “ water droplets could be shifted below -15÷– 20 oC” should be revised as “water droplets could be shifted below -15– 20 oC”. In conclusion section, the future perspective of anti-icing coatings of superhydrophobic surfaces is suggested to provide, especially on the debate issues and the preparation of robust and scalable icephobic coatings.Author Response
Referee 2:
In this review manuscript, the author aimed to reveal the relationship between icephobic and superhydrophobic coatings. Firstly, active and passive anti-icing methods were summarized and their physical mechanisms regulating the ice accretion on solid surfaces were proposed. Then the most important scientific breakthroughs in the field of icephobic coatings were reviewed. The main hypotheses for the anti-icing properties of superhydrophobic surfaces along with their major shortcomings were discussed. Finally, the launch of novel hybrid anti-icing systems and some unknown so far application aspects of icephobic materials in cryobiology were revealed. This review manuscript was well written and the author cited important and comprehensive literatures to support his opinions. I recommend it to be accepted after minor revision. Following are the comments:
The title of the article may be inappropriate, for bio-inspired science or technology were not mentioned much in the manuscript.
Response:
Dr. Esmeryan highly appreciates the positive evaluation of reviewer 2 and his/her very constructive comments and suggestions. He/she is correct that the initial title may be inappropriate. Considering the feedback from reviewers 1 and 2, the author of the review changed the title with such reflecting in a much better way the manuscript’s content.
Change in Manuscript:
The manuscript’s title has been changed.
Why did author use “soot” as key word? Looking through the manuscript, several samples were given using carbon soot, however, the author didn’t explain the functionality of soot.Response:
Thank you very much for this important remark. The reason why the word “soot” appears in the keywords is because it will facilitate the search in the digital databases. Also, comments about the functionality of the soot in the context of anti-icing materials is available on Page 11, Lines 432-434. However, if the reviewer and editors still think this keyword is inappropriate, Dr. Esmeryan would love to remove it during the proofreading stage.
In page 8, line 303, “ water droplets could be shifted below -15÷– 20 oC” should be revised as “water droplets could be shifted below -15– 20 oC”.Response:
Thanks for this kind suggestion. The correction has been done.
Change in Manuscript:
The revised phrase is now available in Page 8, Line 324.
In conclusion section, the future perspective of anti-icing coatings of superhydrophobic surfaces is suggested to provide, especially on the debate issues and the preparation of robust and scalable icephobic coatings.Response:
Dr. Esmeryan agrees with the reviewer. Adding such a discussion in the conclusions section is highly beneficial to the readers.
Change in Manuscript:
Discussion on the durability of icephobic coatings and the existing debate issues is now available in section 8.

Reviewer 3 Report
In this review, the author presents the recent advances in fabricating anti-icing substrates. He mainly focus on whether the superhydrophonibic surfaces have also icephobic characteristics. The above argument is of great interest of the scientific community.
Overall the review is very interesting, well written and I believe should be publish in the Coatings journal.
Author Response
Referee 3
In this review, the author presents the recent advances in fabricating anti-icing substrates. He mainly focus on whether the superhydrophonibic surfaces have also icephobic characteristics. The above argument is of great interest of the scientific community.
Overall the review is very interesting, well written and I believe should be publish in the Coatings journal.
Response:
Dr. Esmeryan is sincerely happy that reviewers 2 and 3 accept positively his way of thinking. Hopefully, if published, the article will gain many pdf downloads and citations, which will be an indication that its content is indeed interesting and attracts the required scientific interest.

Round 2
Reviewer 1 Report
The authors reviewed somehow the manuscript. The title is now more in adequation with the text.
Author Response
Dr. Esmeryan is happy that the reviewer is somehow satisfied with the applied revisions.